# Substantial Effect of Water on Radical Melt Crosslinking and Rheological Properties of Poly(ε-Caprolactone)

**DOI:** 10.3390/polym13040491

**Published:** 2021-02-04

**Authors:** Angelica Avella, Rosica Mincheva, Jean-Marie Raquez, Giada Lo Re

**Affiliations:** 1Department of Industrial and Materials Science, Division of Engineering Materials, Chalmers University of Technology, SE-412 96 Gothenburg, Sweden; avella@chalmers.se; 2Laboratory of Polymeric and Composite Materials (LPCM), Center of Innovation and Research in Materials and Polymers (CIRMAP), University of Mons (UMONS), B-7000 Mons, Belgium; rosica.mincheva@umons.ac.be (R.M.); jean-marie.raquez@umons.ac.be (J.-M.R.)

**Keywords:** reactive melt processing, water-assisted, radical crosslinking, peroxide initiators, biopolymers, poly(ε-caprolactone), rheology

## Abstract

One-step reactive melt processing (REx) via radical reactions was evaluated with the aim of improving the rheological properties of poly(ε-caprolactone) (PCL). In particular, a water-assisted REx was designed under the hypothesis of increasing crosslinking efficiency with water as a low viscous medium in comparison with a slower PCL macroradicals diffusion in the melt state. To assess the effect of dry vs. water-assisted REx on PCL, its structural, thermo-mechanical and rheological properties were investigated. Water-assisted REx resulted in increased PCL gel fraction compared to dry REx (from 1–34%), proving the rationale under the formulated hypothesis. From dynamic mechanical analysis and tensile tests, the crosslink did not significantly affect the PCL mechanical performance. Dynamic rheological measurements showed that higher PCL viscosity was reached with increasing branching/crosslinking and the typical PCL Newtonian behavior was shifting towards a progressively more pronounced shear thinning. A complete transition from viscous- to solid-like PCL melt behavior was recorded, demonstrating that higher melt elasticity can be obtained as a function of gel content by controlled REx. Improvement in rheological properties offers the possibility of broadening PCL melt processability without hindering its recycling by melt processing.

## 1. Introduction

Increased environmental concerns regarding the accumulation of plastic waste in landfills and the marine environment are pushing industry and academia to consider relevant eco-friendly alternatives to conventional plastics [1,2]. Biodegradable thermoplastic polyesters represent a possible solution but generally their processability needs to be enhanced to enable their commercial scaling-up, beyond their economic barriers [3]. Among the commercially available biopolyesters, poly(ε-caprolactone) (PCL) is biodegradable with high ductility and toughness, as well as other mechanical properties comparable with low density polyethylene [4,5]. These characteristics make PCL desirable for various commercial applications like packaging and single-use items, in addition to biomedical applications [6].

However, because of its high linearity, PCL possesses low melt viscosity and strength [7] which limit its processability for example in film blowing, film extrusion or foaming, as seen for other aliphatic biopolyesters [8,9]. Moreover, PCL is characterized by a relatively low crystallization rate [10], reducing the effectiveness of its quenching and injection molding. Several strategies have been proposed to overcome the PCL drawbacks and improve its processability such as blending with other polymer matrices [11,12] and its chemical modification. 

In the latter case, branching and/or crosslinking via ionizing radiation [7,13] or with the use of organic peroxides [13] is a widely explored route to tailor thermo-mechanical and rheological properties of aliphatic biodegradable polyesters such as polylactide and polyhydroxyalkanoates. Moreover, crosslinking PCL has been proven to maintain [14,15] or even favor [16] its biodegradability. In this context, reactive melt processing (REx) is the most employed technique for these chemical modifications in both solvent-free and continuous manners, it does not require any post-purification and eases an industrial uptake [3].

The use of peroxides as radical initiators has been previously reported for PCL backbone modification [17,18] or PCL blends compatibilization [19,20,21]. However, to our knowledge only few works have explored the effect of the only peroxide on PCL branching/crosslinking in solvent casting [15,22] or during melt processing [9,23,24,25]. Gandhi et al. [23] studied the PCL crosslinking with dicumyl peroxide during melt processing, in comparison with radiation crosslinking. The study indicates a more efficient gelation and improvement of PCL rheological properties using peroxide via REx, due to the lower chain scission induced by peroxide compared to radiation. The same system was also presented by Di Maio et al. [9] with the aim of improving PCL viscoelastic properties for foaming. Dynamic rheology of the modified PCL shows an increase in viscosity and melt elasticity with higher amounts of dicumyl peroxide. Structural PCL modification via REx comparing two types of peroxide was investigated by Przybysz et al. [24]. The results point out that different degrees of branching/crosslinking can be achieved as a function of the peroxide structure and amount. Han et al. [25] reported the use of benzoyl peroxide (BPO) to crosslink PCL in a two-step method: first blending at low temperature, followed by crosslinking during compression molding. A maximum in tensile strength was registered at 1 wt.% of BPO, while higher amount of peroxide increased the gel fraction but lowered the mechanical properties. It is worth to note that after a critical gel fraction (≈ 40–50%) the thermoplasticity of the polymer is hampered, together with an increase in its elastic melt behavior, hindering its melt processing and possibility to be melt-recycled. 

Looking into the mechanism of peroxide-induced gelation, peroxide decomposes at high temperatures into free radicals, which then abstract hydrogens directly from the polymer backbone and consequently generate covalent bonds between chains macroradical recombination, yielding branching/crosslinking [26]. It has been demonstrated that radical reaction kinetics can be improved using low viscous media in polyphasic solvent or bulk systems [27]. In the context of REx, the diffusivity of macroradicals can be hindered due to their high viscosity, thus limiting the radical propagation. Therefore, water molecules can be considered as a potential aid in melt radical reactions because of their high mobility. A beneficial effect of water in polymer grafting during free radical graft copolymerization in water suspension has been proven by Kaur et al. [28]. Water-assisted radical reactive melt processing has been successfully exploited for the preparation of cellulose-based PCL nanocomposites [4]. Moreover, water has demonstrated to act as: (i) true catalyst through its ability to form complexes via hydrogen bonds or radical conjugation (catalysis) [29]; (ii) enhancer of the recombination processes (radical complex) [30]; (iii) highly efficient collision partner stabilizing intermediates in radical reactions (energy-transfer) [30]. 

Following these findings, the objectives of our work were the design and the control of PCL crosslinking with low BPO amounts via one-step water-assisted REx to broaden the PCL processability by improving its rheological properties. Low levels in BPO (up to 1 wt.%) were chosen to limit the gelation up to 40%, preserving PCL thermoplasticity and possible melt-recyclability. Our REx design considered the use of water as a temporary low viscosity medium in the melt, to increase the rate of radical propagation, hence the reaction efficiency. The rationale behind the design was to exploit an expected larger diffusivity of water-borne hydroxyl radical compared to PCL macroradicals, before water evaporation. Therefore, a water-assisted processing was carried out at 120 °C in comparison with traditional dry REx. BPO was chosen as peroxide initiator due to its low decomposition temperature [31], consistent with the selected processing temperature. Structural, thermo-mechanical and rheological properties of neat and reacted PCL were analyzed to assess the effectiveness of this strategy. The results showed the formation of higher gel fraction in water-assisted REx, confirming the validity of the formulated hypotheses. PCL melt viscosity increased with the extent of crosslinking (up to 34%), achieving a predominant elasticity of the PCL melt, synonym of improved melt strength and processability [7,32], at the expense of mechanical properties slightly limited by PCL chain scission.

This work provides relevant insights for future controlled water-assisted reactive melt processing of PCL and its composites with hydrophilic reinforcements. Our results deeper explain successful water-assisted REx of cellulose/PCL biocomposites [4], paving the route for the new generation of biodegradable composites with polysaccharides, with improved mechanical and rheological properties.

## 2. Materials and Methods

### 2.1. Materials

Poly(ε-caprolactone) (PCL) Capa6506 was purchased as powder form from Ingevity, (Warrington, UK). According to the supplier, the PCL grade has a mean molecular weight of 50,000 g·mol^−1^, melting temperature (*T*_m_) of 60 °C and a melt flow rate of 7.9–5.9 g/10 min at 190 °C/2.16 kg. Benzoyl peroxide (BPO), under the trade name Luperox A75 (75%, left water) was purchased from Sigma-Aldrich AB (Stockholm, Sweden) and was used without further purification. Dichloromethane (DCM) was purchased from VWR International AB (Stockholm, Sweden) with purity higher than 99.5%.

### 2.2. Reactive Melt Processing

For dry reactive melt processing (REx), PCL powder was manually premixed with different BPO amounts (0–0.1–0.25–0.5 and 1 wt.%)and processed in an internal mixer AEV 330 (50 cm^3^) (Brabender^®^ GmbH & Co., Duisburg, Germany) with counter-rotating screws W50 (feeding for ≈ 5 min at 30 rpm, then 10 min at 60 rpm). The selected processing temperature was 120 °C to enable water evaporation during REx. and at the same time overcome the onset and melting temperatures of BPO (≈98 and 103 °C, respectively [33]). It is worth to note that we assumed the overall 15 min of REx to be sufficient for BPO decomposition, due to its half lifetime at 120 °C of ≈ 3 min [34]. The reacted PCL was coded as PCL-xL, with x indicating the wt.% of BPO present during the reaction. Water-assisted REx was also carried out by premixing PCL with 50 wt.% of deionized water and 0.5 or 1 wt.% BPO. The obtained paste was melt processed under the same conditions of dry REx. It has been assumed a complete water evaporation within the processing time [4]. These two materials were denoted as PCL-0.5Lw and PCL-1Lw, respectively. For further analyses all the reacted PCL samples have been shaped in squared films of 1 mm thickness by compression molding (Buscher-Guyer KHL 100, Zurich, Switzerland) at 120 °C, at 40 bar for 3 min and 500 bar for 1 min.

### 2.3. Characterization Methods

To verify their solubility, neat and reacted PCL samples with 0.5 and 1 wt.% of peroxide during dry and water-assisted REx were dispersed in dichloromethane (0.6 g in 30 mL) and the mixture was magnetically stirred overnight with Teflon coated stir rods.

To separate the soluble and insoluble fractions from boiling dichloromethane (500 mL), a Soxhlet extraction was carried out for 72 h on 5 g samples. The insoluble fraction was filtered off in glass fiber thimbles (Whatman 603G, VWR), dried at room temperature for 48 h and then weighted to measure the gel content of reacted PCL. Finally, the gel fraction was calculated according to Equation (1):(1)$$Gel\;fraction\;\left[\%\right]=\;\frac{w_{i}}{w_{0}}\;\times 100$$
where *w_i_* indicates the weight of the residual insoluble fraction and *w*_0_ indicates the initial weight of the sample.

Size-exclusion chromatography (SEC) was performed in chloroform (CHCl_3_) at 30 °C using an Agilent (Diegem, Belgium) liquid chromatograph equipped with an Agilent degasser, an isocratic HPLC pump (flow rate = 1 mL·min^−1^), an Agilent autosampler (loop volume = 100 μL; solution concentration = 2 mg·mL^−1^), an Agilent-DRI refractive index detector, and three columns: a PL gel 5 μm guard column and two PL gel Mixed-B 5 μm columns (linear columns for separation of molecular weight (PS) ranging from 200 to 4 × 105 g·mol^−1^). Polystyrene standards were used for calibration.

The thermal stability was studied by thermogravimetric analysis (TGA) with a TGA/DSC 3 + Star system (Mettler Toledo, Greifensee, Switzerland). Approximately 5 mg of each sample were preheated from room temperature to 70 °C, where an isothermal segment was maintained for 15 min to remove residual moisture. Then the samples were heated to 550 °C at a heating rate of 5 °C·min^−1^, under N_2_ constant flow of 50 mL·min^−1^. Temperature of the onset of degradation (*T*_5%_), was identified as the temperature at which the weight loss was 5%. Temperature of degradation (*T*_d_) was extrapolated as the temperature of the peak of the first derivative (DTG). Char residue was estimated as the final weight % at 550 °C.

Differential scanning calorimetry (DSC) was performed on a Mettler Toledo DSC 2 calorimeter equipped with a HSS7 sensor and a TC-125MT intercooler. The endotherms were recorded following a heating/cooling/heating temperature profile from −80 °C to 200 °C, at a heating rate of 10 °C·min^−1^, under N_2_ constant flow of 50 mL·min^−1^. The melting temperature (*T*_m_) was detected as the temperature of the maximum of the melting transition peak in the second heating scan, while the glass transition temperature (*T*_g_) at the inflection point of the transition step. Crystallization temperature (*T*_c_) was evaluated as the temperature of the crystallization peak minimum in the cooling scan. The degree of crystallinity (*χ_DSC_)* was calculated according to Equation (2):(2)$$\chi_{DSC}\left[\%\right]=\frac{\Delta H_{M}}{\Delta H_{0}}\times 100$$
where Δ*H_M_* is the specific melting enthalpy and Δ*H_0_* is the melting enthalpy of 100% crystalline PCL (136 J·g^−1^ [4]).

X-Ray diffraction (XRD) spectra were recorded by a D8 Advance Diffractometer (Bruker AXS, Karlsruhe, Germany) with Cr Kα radiation (35 kV, 50 mA) in a 2θ range between 5 and 100°, at a speed of 0.6 deg·min^−1^. The crystallinity of the samples was calculated according to Equation (3):(3)$$\chi_{XRD}\left[\%\right]=\frac{A_{c}}{A_{tot}}\times 100$$
where *A_c_* is the area under the crystalline peaks of the spectra, while *A_tot_* is the total area under the spectra between 2θ = 5 and 100°. The Scherrer equation (Equation (4)) was used to calculate the crystallite size in the direction normal to the 110 lattice planes (*D*_110_):(4)$$D_{110}=\frac{0.9\lambda }{B_{110}\cos\theta }$$
where λ is the radiation wavelength (2.29 Å), *B*_110_ is the full width at half-maximum X-ray diffraction line in radians and θ is the Bragg angle. The 0.9 constant was previously used in literature for similar systems [35,36,37].

Dynamic mechanical properties were evaluated by dynamic mechanical thermal analysis (DMTA) with a DMA Q800 (TA Instruments, New Castle, DE, USA) apparatus in tension-film mode on rectangular bars (25 × 5 × 1 mm^3^). The bars were cut from compression molded films and conditioned for at least 48 h at 23 °C and 53% relative humidity. Temperature ramps were performed from −80 °C to 45 °C at a heating rate of 2 °C·min^−1^, at a frequency of 1 Hz and strain amplitude of 1%, selected in the linear viscoelastic region from strain amplitude sweeps. The glass (*T*_g_) and alpha (*T*_α_) transition temperatures were determined as the temperatures of the maxima of the loss moduli and tanδ, respectively. The damping factor (DF) values were recorded as the maxima of tanδ.

Tensile properties (Young’s modulus, yield stress, ultimate tensile strength, and elongation at break) were carried out according to the standard ASTM D638-14. Dumbbell specimens, with 25 mm gauge length and 1 mm thickness, were cut from compression molded films and conditioned for at least 48 h at 23 °C and 53% relative humidity, prior to testing. At least five specimens were tested for each material at a crosshead speed of 6 mm·min^−1^ with a Zwick/Z2.5 tensile tester (ZwickRoell Ltd., Leominster, UK) equipped with a load cell of 2 kN. Tests were performed and evaluated.

Dynamic rheological measurements were carried out using an Anton Paar MCR 702 rheometer (Graz, Austria) in single-drive mode with a parallel plate geometry (15 mm ø). Disks (20 mm ø) were cut from compression molded films and were conditioned for at least 48 h at 23 °C and 53% relative humidity. The disks were tested at 120 °C and gap of 1 mm, after removal of melt material exceeding the selected geometry. First, oscillatory strain sweep tests were performed in a shear strain range 0.01–100% to determine the linear viscoelastic region. Frequency sweep tests were performed in an angular frequency range from 200 to 0.08 rad·s^−1^ at an applied strain of 1%, within the linear region. Storage modulus (G’), loss modulus (G’’), complex modulus (G*), complex viscosity (η*) and phase angle (δ) were recorded.

## 3. Results and Discussion

### 3.1. Reactive Melt Processing

Chain extension of PCL with a benzoyl peroxide, was successfully carried out via reactive melt processing (REx) in an internal mixer with peroxide content up to 1 wt.% (Figure 1a). During REx no relevant torque increase was detected, indicating that the selected amounts of peroxide did not lead to high crosslinking level, so that the processability of PCL was not hindered. The system was indeed designed to induce a chain extension, i.e., partially crosslinked structure, preserving PCL thermoplasticity. Herein, the purpose of water-assisted REx was to use water as a relatively low viscous phase to favor the PCL radical diffusion [27] and evaluate its catalytic function in radical reactions [29], consequently improving their efficiency.

At relatively high temperature the peroxide decomposes into benzoate (PO*) or phenyl (P*) radicals initiating a free radical mechanism (Figure 1b) [38]. In the water-assisted REx, the interaction of water with the PO* or P* should induce the formation of hydroxyl radical (HO*), as previously reported [29,30,39]. The radicals are then expected to propagate by proton abstraction from the α-carbon relative to the carbonyl group of PCL [17]. Finally, PCL macroradicals can recombine, resulting in branching/crosslinking, or undergo β-scission (Figure 1c) [24,25]. Water can also stabilize the radical structures, suppressing the scission reactions and enhance recombination thus increasing the overall reaction rate [29,30].

To find clear evidence on branching/crosslinking formed during REx, the produced materials have been structurally analyzed.

### 3.2. Structural Analysis

During REx, linear PCL gets first increasingly branched and finally converted into an insoluble gel network [23]. Therefore, its solubility can give an indication about the structure of the reacted PCL and to more extent about the presence of crosslinked chains. In this case, solubility tests have been performed in dichloromethane (DCM), a good solvent for the linear PCL [24]. At a first glance only in the PCL-1Lw dispersion a visible gel was floating, indicating that the radical reaction has taken place during the REx, while all the other samples appeared soluble (Appendix A).

In order to quantify the gel content of the materials, the insoluble fractions were Soxhlet extracted from DCM. It is assumed that the gel contains only the PCL over a critical molecular weight, i.e., 3D crosslinked network and highly branched chains. PCL processed with 1 wt.% peroxide revealed only 1% gel content, in contrast to what found previously by Han et al. [25] that obtained around 40% gel fraction using the same peroxide type and content. However, it is worth to note that they used a higher PCL molecular weight (80,000 g·mol^−1^, from supplier’s datasheet) compared to our study. In this regard, lower molecular weight chains, even if crosslinked, might still be more soluble or have less statistical chance of crosslinking [26], thus leading to lower gel content. This aspect was also evaluated in the work of Navarro et al. [40], illustrating how the gel content of PCL crosslinked by irradiation was strongly dependent on its initial molecular weight.

Instead, in the presence of water at the same peroxide content, a comparable gel fraction was measured (34%). This result confirmed a more efficient crosslinking of PCL structure during water-assisted REx. Similar results on the effect of water were reported for graft free radical copolymerization in a different polymer system [28].

The designed chain extension can be verified by size exclusion chromatography on the soluble fractions. The SEC curves show a monomodal distribution (Figure 2) and are broadened after REx. The reaction of polyesters with peroxide can lead to two reaction products: branched/crosslinked chains and β-scission [24,25]. REx with 0.5 wt.% of peroxide resulted in higher weight-average molecular weight ($\bar{M_{w}}$) and dispersity (Ð), compared to PCL (Table 1), highlighting a successful branching/crosslinking reaction. However, from the curve an increased population of low molecular weight (M_w_) chains is visible, resulting from the β-scission, and contributing to the increased dispersity. The presence of water results in a further increased $\bar{M_{w}}$ and dispersity, but slightly lower amount of low M_w_ chains. Water strongly improves the reaction efficiency, leading to higher branching/crosslinking, compared to the competing β-scission. This is consistent with the gel content results (at 1 wt.% peroxide). It is worth to note that large presence of water (>50 wt.%) in PCL melt processing does not significantly decrease its molecular weight due to water-induced hydrolysis [41].

### 3.3. Thermal Properties

Reduced thermal stability and a decreased onset of degradation are generally indicative of any degradative phenomena or chain scission and can be assessed by thermogravimetric analysis [42].

The thermograms show a reduced degradation onset of PCL, that is particularly evident at low amount of peroxide (<0.5 wt.%) (Table 2 and Appendix A). This result suggests that PCL chain scission is prevalent at lower amount of peroxide (<0.5 wt.%), while branching/crosslinking increases at larger amounts (0.5 and 1 wt.%) [42]. However, the degradation behavior and temperature of PCL have no significant variation after REx with peroxide (Appendix A and Figure 3).

Structural changes can be reflected on the PCL transition temperatures and crystallinity, so the materials were analyzed by DSC and XRD.

The *T*_m_ of the materials is not affected by REx (Appendix A) [43] while noticeable differences are observed in the DSC cooling scans in which a shift of the *T*_c_ to higher values is recorded with increasing the peroxide amount (1 wt.%) (Figure 3 and Table 2) [24,25,44]. This effect can be ascribed to low M_w_ fractions that facilitate the crystallization process due to their higher mobility and easier folding [43,45]. However, it is worth to note that our results can also be explained by a heterogeneous nucleation effect of the branching/crosslinking points, already observed in similar systems [46,47]. Moreover, the crystallization peaks are broadened with increasing amount of peroxide. According to SEC analysis, PCL dispersity increased after REx due to the formation of fractions both at higher and lower M_w_. The increased dispersity can result in more heterogeneous spherulite species, leading to broader crystallization [45,48]. In presence of water the crystallization process is completed at higher temperatures and the increase of the *T*_c_ is more significant, in agreement with its higher gel content measured.

For a better understanding of PCL crystalline features after REx, XRD spectra have been collected on representative samples at higher gel content (1 wt.% peroxide). The diffraction spectra show two main crystalline peaks around 32.1° and 35.6°, previously assigned to (110) and (200) crystallographic lattices [36], which are not shifted in the reacted PCL (Appendix A). The ratio between the area of crystalline peaks and the total area of the spectrum provides indications on the crystallinity degree. Compared to neat PCL, the calculated crystallinity degree slightly increases by ≈ 10% after reaction with 1 wt.% peroxide (Table 2). Increase in crystallinity has been similarly reported in other crosslinked systems [40,47,49], explained by a nucleating effect of the crosslinked network. Additionally, to verify if REx led to different crystalline structures, Scherrer equation was used to calculate the crystallite size. The results indicate an increased crystallite size in the direction normal to the 110 lattice after REx, conceivably due to steric effects induced by PCL branching points. Both crystallinity degree and crystallite size are further enhanced by the presence of water, in line with the larger branching/crosslinking achieved. Mishra et al. [45,50] also recorded similar increases in PCL crystallinity and crystallite size in crosslinked PCL blends by XRD analysis. Nevertheless, hereafter reported bulk properties of modified PCLs cannot be inferred by the small changes observed in their crystallinity. To correlate PCL structural and thermal properties to its performance, dynamic mechanical and tensile behaviors were investigated. Moreover, DMTA can provide more accurate information than DSC on the transition temperatures, *e.g.*, being the glass transition a first order transition in the loss moduli.

### 3.4. Mechanical Properties

The curves (Figure 4) from DMTA in tensile mode indicate that only one transition occurs for neat and reacted PCL. Both the glass (*T*_g_) and alpha (*T*_α_) transition temperatures (Table 3) show a slight shift towards higher temperatures with increasing amount of peroxide. 

The delay in these transitions further demonstrates the formation of branching/crosslinking during REx, which hinder the macromolecular mobility [51]. Indeed, the largest shift in transition temperatures is reported for PCL-1Lw, according to its higher gel content achieved. Only for this material, the storage modulus improves for a large range of temperatures (from −70 to −20 °C) (Figure 4). For all the other materials, no reinforcement is shown due to the low gel content and the plasticizing effect of the low M_w_ chains.

It is worth noting that limited reinforcement of crosslinks is expected after the glass transition, as previously observed in similar systems [24]. However, PCL damping factor decreases with increasing peroxide amount, indicating a more elastic behavior of the polymer after REx. The limited reinforcement above the *T*_g_ observed from DMTA is also reflected in the tensile properties of the materials recorded at room temperature (Figure 5 and Table 4). The Young’s modulus and yield stress of PCL are slightly lowered after REx and further decreased when the reaction was carried out in the presence of water, while PCL elongation and strength are preserved up to 0.5 wt.% in peroxide (Appendix A and Appendix A). Water-assisted REx with 1 wt.% led to a ≈ 15% increase of ultimate tensile strength compared to unmodified PCL, resulting from increased strain hardening, and a slight improvement of elongation compared to PCL-1L. Similar reduction of PCL tensile properties after crosslinking has been previously reported [25], with only exception of a small increase in PCL strength, as also achieved in this work.

After analyzing the thermomechanical properties below the glass transition and at room temperature, parallel plate rheology was used to explore the behavior of the material in the melt state, crucial for investigating changes in PCL melt processibility. Moreover, rheological analysis allows further understanding of the macromolecular structure.

### 3.5. Dynamic Rheology

From the isothermal frequency sweeps at 120 °C (Figure 6a), a substantial improvement of the shear storage modulus of PCL up to four orders of magnitude is observed with increasing crosslinking level and progressively reduced slopes correspond to higher moduli [45]. Even if less pronounced, the loss modulus also increased as a consequence of the increased $\bar{M_{w}}$ [23]. At higher level of peroxide, PCL melt presented a transition from viscous to elastic behavior. A crossover point around 6 rad·s^−1^ of frequency is observed for PCL-1L, while a complete inversion is recorded for PCL-1Lw. The prevalent elastic melt behavior in the entire frequencies range is inferred by increased chain entanglement for the high molecular weight crosslinked PCL [44].

Viscous to elastic transition has been highlighted in the Van Gurp-Palmen plot (Figure 6c), where the recorded below 45° phase angles point out a predominant elasticity of the melt [5]. The Van Gurp-Palmen plot of PCL-1Lw shows the lowest phase angles (<45°), confirming the predominant elastic behavior of the higher crosslinked PCL melt, as illustrated in the figure. Instead, the phase angles recorded for the unmodified PCL are close to 90°, indicating a typical flow behavior of viscoelastic fluid with highly linear structure and low dispersity [47,52]. Prevalent branched structure of PCL is assumed for intermediate gel content and phase angles between 70 and 45° (Figure 6c). The commented rheological transition liquid to solid-like is also reflected in a dramatic increase in viscosity. The complex viscosity of PCL shows a typical Newtonian plateau followed by a minimal shear thinning region at higher frequencies (Figure 6b). The complex viscosity of the modified PCL shows a non-Newtonian behavior, with pronounced shear thinning that increases with the crosslinking level [8]. Accordingly, the shear thinning for water-assisted REx materials it is the more pronounced. Up to two orders of magnitude increase is measured for complex viscosity of the modified PCL at low frequencies, in agreement with the increased $\bar{M_{w}}$ and higher entanglement of the branched structure [8,49]. At higher frequencies the impact of peroxide-initiated reactions is lower as the viscosities converge (at frequencies around 200 rad·s^−1^). It is worth to note the substantial effect of water can be already observed at lower level of peroxide. PCL-0.5Lw showed higher resistance against high shear forces than the neat and modified PCL via dry REx.

Accordingly, increase in melt viscosity and shear moduli suggests that the melt strength of PCL can be controlled by the only reaction with peroxide, thus improving its processability [44].

## 4. Conclusions

Despite the extensive use of peroxide as crosslinking agent, this study showed how water-assisted REx can be employed with low levels of peroxide to control PCL crosslinking and improve its melt elasticity while preserving its thermoplasticity. Water was chosen as catalyst for the designed Rex under the hypothesis that a low viscosity medium can boost the radical reaction by increasing the radical diffusion. The results showed that the presence of water increased PCL molecular weight and gel content compared to the dry process, from 1% to 34% with 1 wt.% peroxide, confirming a more efficient radical propagation in water-assisted REx. Differential scanning calorimetry showed increased crystallization temperatures and easier crystallization process of reacted PCL, compared to neat PCL. From the dynamic mechanical analysis, higher branching/crosslinking slightly increased the transition temperatures and led to a reinforcement effect in the temperature range below the glass transition. After the glass transition, the mechanical reinforcement was limited, both in dynamic mechanical and tensile properties, reflecting the plasticizing effect of low molecular weight chains formed by PCL β-scission during peroxide-initiated reactions. Instead, in the melt state the effect of branching/crosslinking was more evident as shown by melt rheology. The rheological behavior of crosslinked PCL showed a transition from the typical viscous-like to solid-like. The shear storage moduli were increased by the reactive melt processing, confirming the desired improvement of PCL rheological properties. Increasing the gel content, the viscosity of reacted PCL increased at low frequencies, while higher shear thinning behavior was shown in the entire frequency range. At higher frequencies, approaching typical melt processing conditions, viscosity of crosslinked PCL converged to the neat biopolyesters values, underlying that a controlled water-assisted REx can improve PCL rheological properties without affecting its processability. This work represents a relevant reference for future controlled water-assisted reactive melt processing of polymers and composites, in which water could have the role of feeding medium, e.g., for polysaccharides suspension, but also useful to boost peroxide-initiated reactions.

## Figures and Tables

**Figure 1 polymers-13-00491-f001:**
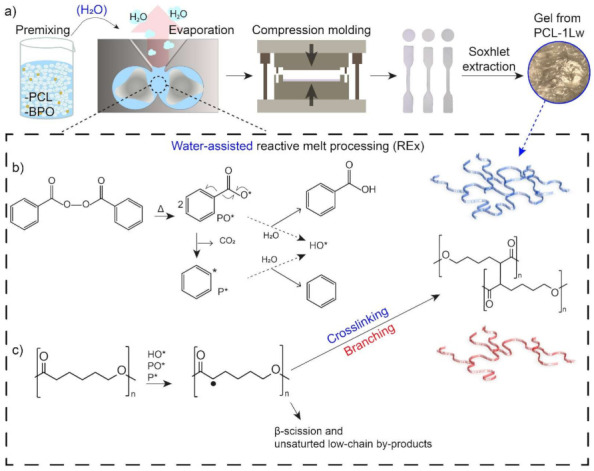
(**a**) Scheme of sequential steps: REx with or without water, compression molding of test specimens and Soxhlet extraction to recover gel fractions. The photo refers to insoluble fraction of PCL reacted with 1 wt.% peroxide during water-assisted REx. Schematics of (**b**) peroxide activation and reactions with water and (**c**) reaction products.

**Figure 2 polymers-13-00491-f002:**
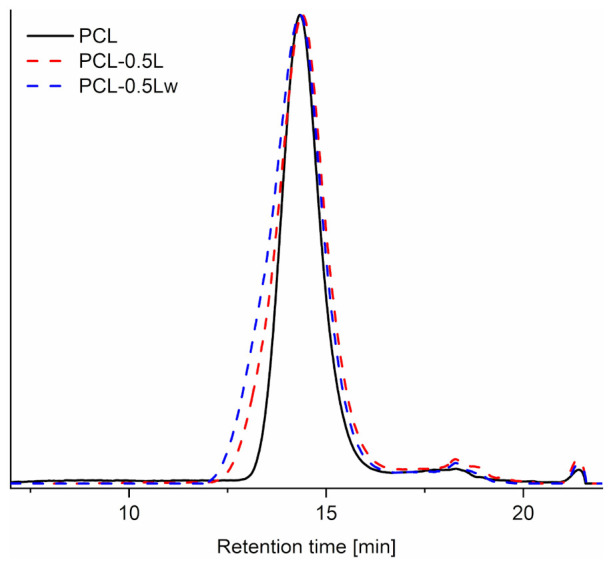
SEC chromatograms of fractions completely soluble in chloroform of PCL and PCL reacted with 0.5 wt.% peroxide during dry (PCL-0.5L) or water-assisted (PCL-0.5Lw) REx.

**Figure 3 polymers-13-00491-f003:**
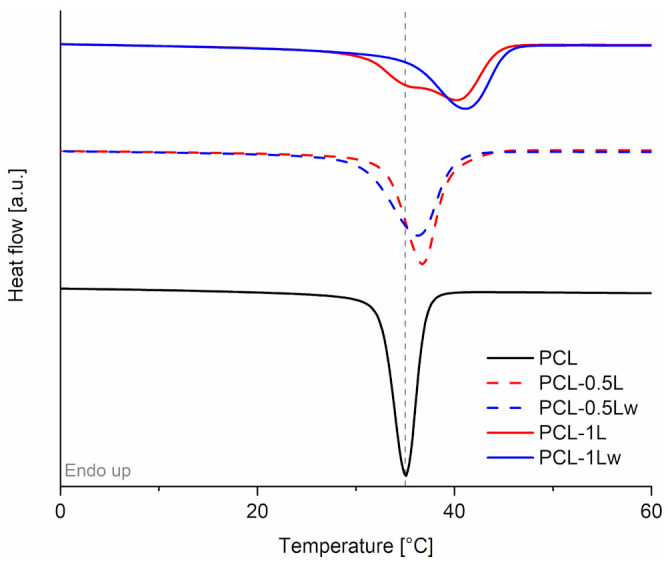
DSC cooling scan of neat and reacted PCL.

**Figure 4 polymers-13-00491-f004:**
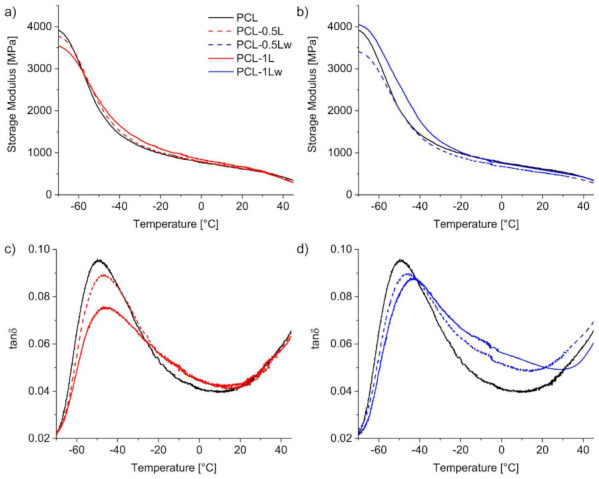
Representative curves of storage moduli (**a**), (**b**) and tanδ (**c**),(**d**) recorded from DMTA temperature sweep of PCL and reacted PCL during dry (**a**),(**c**) and water-assisted (**b**), (**d**) REx.

**Figure 5 polymers-13-00491-f005:**
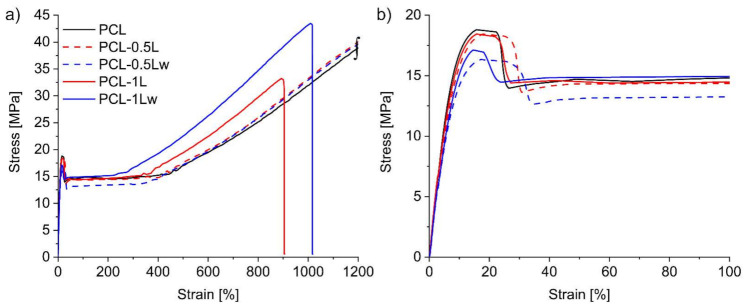
(**a**) Stress-strain curves from tensile tests at room temperature of neat and reacted PCL and (**b**) their magnification on the yield region.

**Figure 6 polymers-13-00491-f006:**
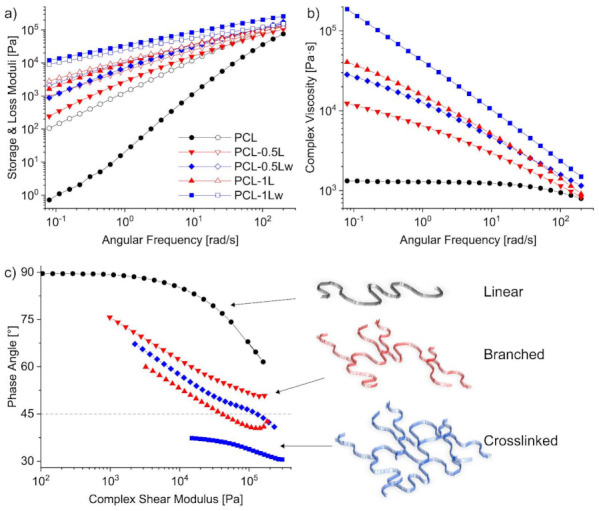
Dynamic rheological behavior in frequency sweeps at 120 °C of neat and reacted PCL: (**a**) storage (full symbols) and loss (empty symbols) moduli; (**b**) complex viscosity; (**c**) Van Gurp-Palmen plot showing phase angles as a function of complex modulus, phase angles below 45° indicate predominant elasticity.

**Table 1 polymers-13-00491-t001:** Number ($\bar{M_{n}})$ and weight $\left(\bar{M_{w}}\right)\;$ average molecular weights rounded off and dispersity (Ð) detected from SEC of fractions completely soluble in chloroform of PCL and PCL reacted with 0.5 wt.% peroxide during dry (PCL-0.5L) or water-assisted (PCL-0.5Lw) REx.

Material	$\bar{M_{n}}\;(g\cdot {\mathrm{mol}}^{-1})$	$\bar{M_{w}}\;(g\cdot {\mathrm{mol}}^{-1})$	Ð
PCL	77,000	150,000	2
PCL-0.5L	74,000	220,000	2.92
PCL-0.5Lw	86,000	290,000	3.37

**Table 2 polymers-13-00491-t002:** Thermal properties of neat and reacted PCL detected by TGA, DSC and XRD analyses. Onset temperatures of degradation (*T*_5%_) evaluated at 5% weight losses in TGA. Crystallization temperatures (*T*_c_) detected from the DSC cooling scans. Crystallinity (χ_XRD_) and crystal size in the direction perpendicular to (110) lattice plane (D_110_) calculated from XRD diffractograms.

Material	*T*_5%_ (°C)	*T*_c_ (°C)	χ_XRD_ (%)	D_110_ (Å)
PCL	363	35.1	38.2	223
PCL-0.5L	331	36.7	-	-
PCL-0.5Lw	325	36.4	-	-
PCL-1L	345	35.5–40.2	41.4	260
PCL-1Lw	347	41.2	42.8	291

**Table 3 polymers-13-00491-t003:** DMTA main results of neat and reacted PCL recorded by DMTA Glass transition temperature (*T*_g_) as temperature of loss modulus peak maximum. Alpha transition temperature (*T*_α_) as temperature of tanδ peak maximum. Damping factor (DF) as the maximum of tanδ. Storage modulus (E’) values at −70 and 20 °C. Scattering of the data below 3%.

Material	*T*_g_ (°C)	*T*_α_ (°C)	DF	E’ _−70 °C_ (MPa)	E’_20 °C_ (MPa)
PCL	−56.2	−49.4	0.095	3920	613
PCL-0.5L	−54.8	−46.6	0.089	3770	621
PCL-0.5Lw	−54.4	−45.7	0.090	3405	528
PCL-1L	−54.0	−45.7	0.075	3544	669
PCL-1Lw	−52.1	−42.9	0.088	4051	591

**Table 4 polymers-13-00491-t004:** Mechanical properties assessed from tensile tests at room temperature. Each value represents the average of 5 measurements with the standard deviation. * Data extracted at the upper limit of the instrument; break did not occur.

Material	Young’s Modulus (MPa)	Yield Stress (MPa)	Ultimate Tensile Strength (MPa)	Elongation at Break (%)
PCL	284 ± 5	19.2 ± 0.5	> 38.7 *	> 1200 *
PCL-0.5L	248 ± 6	18.4 ± 0.1	> 38.8 *	> 1175 *
PCL-0.5Lw	218 ± 8	16.0 ± 0.4	> 39.8 *	> 1200 *
PCL-1L	242 ± 13	17.3 ± 1	33.6 ± 1.3	941 ± 57
PCL-1Lw	225 ± 14	17.1 ± 0.5	44 ± 1.2	1035 ± 48

## Data Availability

The data presented in this study are available on request from the corresponding author.

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
