# Peer review of "Substantial Effect of Water on Radical Melt Crosslinking and Rheological Properties of Poly(ε-Caprolactone)"

_polymers, 2021, doi:10.3390/polym13040491_

Round 1
Reviewer 1 Report
The manuscript entitled "Substantial effect of water on radical melt crosslinking and rheological properties of poly(ε-caprolactone)" is an interesting study in which authors have reported their findings on using one strategy for design and processing of PCL and its composites with improved properties. Having in mind the range of applications of these polymers, the data presented in this paper provide useful information for future researches, paving the route for the new generation of biodegradable composites.
This study is well-designed and organized, also results are clearly presented and nicely written with scientific explanations and references. Moreover, Results and discussion section is well represented by good quality of pictures. Conclusions are well written, providing needed information.
Author Response
Point-by-point response to the reviewers’ comments
Reviewer 1
Open Review
(x) I would not like to sign my review report
( ) I would like to sign my review report
English language and style
( ) Extensive editing of English language and style required
(x) Moderate English changes required
( ) English language and style are fine/minor spell check required
( ) I don't feel qualified to judge about the English language and style
|
Yes |
Can be improved |
Must be improved |
Not applicable |
|
|
Does the introduction provide sufficient background and include all relevant references? |
(x) |
( ) |
( ) |
( ) |
|
Is the research design appropriate? |
(x) |
( ) |
( ) |
( ) |
|
Are the methods adequately described? |
(x) |
( ) |
( ) |
( ) |
|
Are the results clearly presented? |
(x) |
( ) |
( ) |
( ) |
|
Are the conclusions supported by the results? |
(x) |
( ) |
( ) |
( ) |
Comments and Suggestions for Authors
The manuscript entitled "Substantial effect of water on radical melt crosslinking and rheological properties of poly(ε-caprolactone)" is an interesting study in which authors have reported their findings on using one strategy for design and processing of PCL and its composites with improved properties. Having in mind the range of applications of these polymers, the data presented in this paper provide useful information for future researches, paving the route for the new generation of biodegradable composites.
This study is well-designed and organized, also results are clearly presented and nicely written with scientific explanations and references. Moreover, Results and discussion section is well represented by good quality of pictures. Conclusions are well written, providing needed information.
Submission Date
21 January 2021
Date of this review
28 Jan 2021 14:53:58
We thank the Reviewer for his opinion and for considering our study useful for future research. We are grateful for his recognition of our study design, organization, results and discussion presentation, and to find adequately supported our conclusions. We have taken the chance to improve the manuscript according to all the Reviewer' suggestions to improve the English language.
Improvements in the English language have been made according to the Reviewer comments. In particular, the changes have been made at the following lines (l.):
l.54: “that” has been changed with “, it”
l.85: “et. al. [28]” has been changed with “et al. [28].”
l.230: “evidences” has been changed with “evidence” (l.247)
l.243: “wt%” is now “wt.%” (l.260)
l.255: “A successful chain extension designed via REx with peroxide can be assessed also by SEC “ has been changed in “The designed chain extension can be verified by size exclusion chromatography” (l-272)
l.274: “wt%” is now “wt.%” (l.289 and in the overall manuscript)
l.276: “A reduced” has been replaced by “Reduced”; and “thermo-gravimetrical” by “thermo-gravimetric” (l.295-297)
l.323: “Nevertheless, the observed changes in crystallinity are not significant enough to affect PCL properties analyzed hereafter.” Has been replaced by “Nevertheless, hereafter reported bulk properties of modified PCLs cannot be inferred by the small changes observed in their crystallinity.2 (l.341-343)
l.325-327: The period was revised in “To correlate PCL structural and thermal properties to its performance, dynamic mechanical and tensile behaviors were investigated. Moreover, DMTA can provide more accurate information than DSC on the transition temperatures, e.g., being the glass transition a first order transition in the loss moduli.” (l.345-348)
l.370-372: The sentence has been changed with “After analyzing the thermomechanical properties below the glass transition and at room temperature, parallel plate rheology was used to explore the behavior of the material in the melt state, crucial for investigating changes in PCL melt processibility.” (l.392-394)
l.381 and l.390: The “Dynamic rheology” section has been thoroughly revised, for the sake of improved clarity.
l.430: The sentence was changed with “The shear storage moduli were increased by the reactive melt processing, confirming the desired improvement of PCL rheological properties.” (l.454-456)
The sentence “The shear storage moduli were increased by REx and were predominant in all the frequency range investigated, indicating a strong improvement of PCL rheological properties in water-assisted REx with peroxide.” has been changed in “Differential scanning calorimetry showed increased crystallization temperatures and easier crystallization process of reacted PCL, compared to neat PCL.” (l.465-466)

Reviewer 2 Report
The authors address the attractive topic. The materials taken into account are compared thoroughly and the experimental results justify the authors' conclusions. The manuscript is well arranged with suitable referencing. However, a couple of items should be presented in more detail:
- The producers (ll. 116, 120, 121, 126, 156, 163, 173, 183, 195, 197, etc.) should be more specified.
- l. 140: magnetically stirred, this preparation participates – to some extent – in material degradation. The authors should describe this process in more detail including the process parameters + cross or rod, teflon coated?
- l. 197: The twin-drive rheometer MCR 702 was used as a twin-drive or a single-drive rheometer?
- Table 1 Are the M values rounded off?
- Table 2: As the values Tc are relatively close, it should be better to introduce 1 decimal figure.
- Table 3: For Tg, Ta as in Table 2.
- Fig. 6b: It seems that PCL-0.5Lw is more resistant against higher hydrodynamic forces than the remaining 4 materials. Is there any reason for that?
Just for improvement:
- l. 245: to delete C.
- l. 266: consistent
- l. 332: occurs
Author Response
Point-by-point response to the reviewers’ comments
Reviewer 2
Open Review
(x) I would not like to sign my review report
( ) I would like to sign my review report
English language and style
( ) Extensive editing of English language and style required
( ) Moderate English changes required
( ) English language and style are fine/minor spell check required
(x) I don't feel qualified to judge about the English language and style
|
Yes |
Can be improved |
Must be improved |
Not applicable |
|
|
Does the introduction provide sufficient background and include all relevant references? |
(x) |
( ) |
( ) |
( ) |
|
Is the research design appropriate? |
(x) |
( ) |
( ) |
( ) |
|
Are the methods adequately described? |
( ) |
(x) |
( ) |
( ) |
|
Are the results clearly presented? |
( ) |
(x) |
( ) |
( ) |
|
Are the conclusions supported by the results? |
(x) |
( ) |
( ) |
( ) |
Comments and Suggestions for Authors
The authors address the attractive topic. The materials taken into account are compared thoroughly and the experimental results justify the authors' conclusions. The manuscript is well arranged with suitable referencing. However, a couple of items should be presented in more detail:
We are grateful for the overall positive comment expressed on our study and our manuscript.
- The producers (ll. 116, 120, 121, 126, 156, 163, 173, 183, 195, 197, etc.) should be more specified.
We thank the Reviewer to bring to our attention this lacking aspect of our manuscript. In the revised methods section, specifications have been added regarding the producers of materials and instruments used. Moreover, the description of the methods was reviewed, for the sake of clarity.
- l. 140: magnetically stirred, this preparation participates – to some extent – in material degradation. The authors should describe this process in more detail including the process parameters + cross or rod, teflon coated?
l.140-145 the method of magnetic stirring description has been improved accordingly to the reviewer’s comment, and more details have been added.
- l. 197: The twin-drive rheometer MCR 702 was used as a twin-drive or a single-drive rheometer?
The use of a single-drive rheometer has been specified (l.206)
- Table 1 Are the M values rounded off?
We thank the Reviewer for the valuable comment. In the revised Table 1, the caption has been specified by adding that the molecular weight values are rounded off.
- Table 2: As the values Tc are relatively close, it should be better to introduce 1 decimal figure.
In Table 2: a decimal figure has been added to the crystallization temperatures, according to the Reviewer’s suggestion.
- Table 3: For Tg, Ta as in Table 2.
In the revised Table 3, a decimal figure has been added to the transition temperatures
- Fig. 6b: It seems that PCL-0.5Lw is more resistant against higher hydrodynamic forces than the remaining 4 materials. Is there any reason for that?
In the revised manuscript, this aspect has been discussed and it has offered a further opportunity to highlight the effect of water during REx. We thank the Reviewer to have brought to our attention this rheological feature for the water-assisted REx processed material with low content of radical initiator. This has led us to further improving our rheological analysis, so the manuscript quality.
Just for improvement:
- l. 245: to delete C.
“C.” has been removed (l.244)
- l. 266: consistent
The word “consist” has been modified to “consistent” (l.265)
- l. 332: occurs
The word “occurrs” has been modified to “occurs”( l.331)
Submission Date
21 January 2021
Date of this review
27 Jan 2021 22:19:41

Reviewer 3 Report
By designing a water assisted Rex, the authors compare it with the assumption that the free radical diffusion of PCL in molten state is slower. The effect of drying and water assisted Rex on PCL, its structure, thermomechanical properties and rheological properties were studied. The work design is reasonable and the results are reliable, which has a certain guiding significance for polymer processing and manufacturing.
Nevertheless, I found that in some parts of the article, the authors had a presupposed position on the experiment, and some of the descriptions were not objective enough, so I suggest that it should be carefully revised. In addition, the errors in Table 4 should be unified, especially for EB. I hope to review a more objective and readable manuscript which can be published.
Author Response
Point-by-point response to the reviewers’ comments
Reviewer 3
Open Review
(x) I would not like to sign my review report
( ) I would like to sign my review report
English language and style
( ) Extensive editing of English language and style required
( ) Moderate English changes required
( ) English language and style are fine/minor spell check required
(x) I don't feel qualified to judge about the English language and style
|
Yes |
Can be improved |
Must be improved |
Not applicable |
|
|
Does the introduction provide sufficient background and include all relevant references? |
(x) |
( ) |
( ) |
( ) |
|
Is the research design appropriate? |
(x) |
( ) |
( ) |
( ) |
|
Are the methods adequately described? |
(x) |
( ) |
( ) |
( ) |
|
Are the results clearly presented? |
( ) |
( ) |
(x) |
( ) |
|
Are the conclusions supported by the results? |
( ) |
(x) |
( ) |
( ) |
Comments and Suggestions for Authors
By designing a water assisted Rex, the authors compare it with the assumption that the free radical diffusion of PCL in molten state is slower. The effect of drying and water assisted Rex on PCL, its structure, thermomechanical properties and rheological properties were studied. The work design is reasonable and the results are reliable, which has a certain guiding significance for polymer processing and manufacturing.
We thank the Reviewer for his overall positive evaluation of our work and for the recognition of its significance for the topics of polymer processing and manufacturing, which indeed are the targeted audience of the study.
Nevertheless, I found that in some parts of the article, the authors had a presupposed position on the experiment, and some of the descriptions were not objective enough, so I suggest that it should be carefully revised. In addition, the errors in Table 4 should be unified, especially for EB. I hope to review a more objective and readable manuscript which can be published.
Submission Date
21 January 2021
Date of this review
25 Jan 2021 10:51:09
As suggested by the Reviewer, some of the descriptions and discussions have been revised, for the sake of overall clarity. In Table 4, the table heading has been modified, specifying that “break did not occur” and the errors reported have been adjusted accordingly.
We believe that the objectivity, therefore the quality of the revised manuscript has been further improved, thanks to the review process raised up by the valuable comments of the Reviewers.
